



# Incorporation of pollen data in source maps is vital for pollen dispersion models

Alexander Kurganskiy[1,2,3], Carsten Ambelas Skjøth[3], Alexander Baklanov[4], Mikhail Sofiev[5], Annika Saarto[6], Elena Severova[7], Sergej Smyshlyaev[2], and Eigil Kaas[1]

[1]Niels Bohr Institute, University of Copenhagen, Copenhagen, Denmark
[2]Russian State Hydrometeorological University, St. Petersburg, Russia
[3]School of Science and the Environment, University of Worcester, Worcester, United Kingdom
[4]World Meteorological Organization, Geneva, Switzerland
[5]Finnish Meteorological Institute, Helsinki, Finland
[6]Biodiversity Unit of the University of Turku, Turku, Finland
[7]Lomonosov Moscow State University, Moscow, Russia

**Correspondence:** Alexander Kurganskiy (a.kurganskiy@worc.ac.uk)

**Abstract.** Information about distribution of pollen sources, i.e. their presence and abundance in a specific region, is important especially when atmospheric transport models are applied to forecast pollen concentrations. The goal of this study is to evaluate three pollen source maps using an atmospheric transport model and study the effect on the model results by combining these source maps with pollen data. Here we evaluate three maps for the birch taxon: (1) a map derived by combining land cover data and forest inventory; (2) a map obtained from land cover data and calibrated using model simulations and pollen observations; (3) a statistical map resulting from analysis of forest inventory and forest plot data. The maps were introduced to the Enviro-HIRLAM (Environment – High Resolution Limited Area Model) as input data to simulate birch pollen concentrations over Europe for the birch pollen season 2006. 18 model runs were performed using each of the selected maps in turn with and without calibration with observed pollen data from 2006. The model results were compared with the pollen observation data at 12 measurement sites located in Finland, Denmark and Russia. We show that calibration of the maps using pollen observations significantly improved the model performance for all three maps. The findings also indicate the large sensitivity of the model results to the source maps and agree well with other studies on birch showing that pollen or hybrid-based source maps provide the best model performance. This study highlights the importance of including pollen data in the production of source maps for pollen dispersion modelling and for exposure studies.

## 1 Introduction

Aeroallergens are a specific type of atmospheric aerosols causing allergic reactions among people suffering from allergic rhinitis and it is often connected with asthma (Bachert et al., 2004). The amount of allergic patients sensitive to pollen is assessed





to be 20 % of the European population (WHO, 2003). There are substantial variations in sensitization levels towards specific aeroallergens in Europe. (Heinzerling et al., 2009). In Northern Europe, pollen from Poaceae (grasses) and Betulaceae (e.g. hazel, alder and birch) families show the highest sensitization among patients (Heinzerling et al., 2009). Birch pollen is generally the most abundant pollen type of those four pollen types (Skjøth et al., 2013b), often with large interannual fluctuations
in the overall pollen integral (e.g., Piotrowska and Kaszewski, 2011). In particular, for Northern Europe the pollen season of 2006 was characterized by abundant birch pollen concentrations that resulted in an increase of patient calls for medical assistance during the spring period (Sofiev et al., 2011). Such episodes can potentially be described and forecast using atmospheric dispersion models (Sofiev et al., 2006) helping sensitized individuals to minimize their symptoms. Several European countries, such as Denmark, have seen an increase in sensitized individuals over the past few decades (e.g., Rasmussen, 2002; Linneberg
et al., 2007). Generally, skin prick test (SPT) is performed to identify sensitization of patients to common pollen allergens (e.g., Heinzerling et al., 2009). Haahtela et al. (2015) showed that the percentage of the population with positive SPT was about 35 % and 20 % in Finish and Russian Karelia, respectively. These facts demonstrate the importance of pollen studies using atmospheric dispersion models applied for research and operational purposes (Klein et al., 2012).

A variety of pollen types are considered as biological components by several atmospheric dispersion models such as SILAM,
COSMO-ART, WRF/CMAQ, Enviro-HIRLAM and others (see e.g., Zink et al., 2012, 2013; Zhang et al., 2014; Sofiev et al., 2015a; Baklanov et al., 2017; Sofiev, 2017; Zink et al., 2017). Information about distribution of pollen sources, i.e. their presence and abundance in a region of study, plays a crucial role in aerobiological modelling and forecasting using atmospheric dispersion models (e.g., Zink et al., 2017). In other words, it is one of the key input data used to simulate or forecast pollen emission, its atmospheric transport and depositions in these models. For this reason, recent studies (e.g., Pauling et al., 2012;
Bonini et al., 2018) are devoted to developing pollen source inventory/maps. Two main approaches aiming to build/construct the pollen source inventories are suggested in the literature: bottom-up and top-down (Skjøth et al., 2013b). The bottom-up approach is based on statistical data of pollen source distribution combined with land cover data obtained from remote sensing (Skjøth et al., 2008). The limitation of this approach is that statistical data are either unavailable or do cover too large territory (e.g. the whole country) in some regions. The top-down approach uses pollen observations as starting point to estimate the
abundance of different pollen taxa and it can be combined with land use data and/or models for more detailed assessment of pollen sources on a regional scale. Poor geographical and temporal coverage of the pollen observation data can limit this approach in some regions.

Skjøth et al. (2008) used the bottom-up approach to create the Tree Species Inventory (TSI) for 39 species (including birch, oak, alder etc.) redistributed to a 50 km grid to be used in atmospheric dispersion models. The top-down approach has been
used to produce ragweed pollen inventories for the Pannonian Basin (Skjøth et al., 2010), France (Thibaudon et al., 2014), Austria (Karrer et al., 2015) and Italy (Bonini et al., 2018). The inventories are based on combination of pollen observation data, knowledge on ragweed ecology and detailed land cover information. The latter one (i.e., Bonini et al., 2018) also includes influence of the Ophraella communa beetle on the plants. Prank et al. (2013) as well as Hamaoui-Laguel et al. (2015) have both used combined occupancy and climatic habitat quality maps for ragweed from an ecological model. The map covering
Europe has been introduced to the atmospheric models as input data, run for several years and calibrated using pollen obser-





vation data. A birch pollen source map has been derived by Pauling et al. (2012). The methodology includes using a forest inventory combined with local and global land use datasets. The obtained map has been calibrated by observational pollen data and it covers Europe. An urban-scale grass pollen source inventory has been obtained by Skjøth et al. (2013a) using GIS and remote sensing data. A high resolution (1 km) pollen source inventory based on 12 taxa of trees, grass and weeds has been

produced by McInnes et al. (2017) for the UK to be mainly used in exposure studies as well as other possible applications. Many of these different pollen source maps show substantial variation, exemplified by the birch maps over the UK produced by McInnes et al. (2017) and Pauling et al. (2012) or the European scale by comparing the maps by Pauling et al. (2012) with Siljamo et al. (2013). This suggests substantial disagreement or uncertainty in the production of the pollen source maps – the key input data for pollen dispersion models. An additional problem is the large variability of the pollen production per plant in

each specific year. Several approaches have been proposed to address this issue (e.g., Masaka and Maguchi, 2001; Ranta et al., 2005; Ritenberga et al., 2018).

Zink et al. (2017) applied an atmospheric dispersion model to compare different ragweed pollen source maps using the COSMO-ART model. The results show that the pollen source map resulting from combining land cover and pollen observation data provides the best model performance. The advantage of including a dispersion model is that, in principle, such

models take into account atmospheric transport and its effect on pollen concentrations during a specific year (e.g., Prank et al., 2013). A recent study showed high potential of extended 4-dimensional variational data assimilation as a rigorous procedure for obtaining the source maps calibrations (Sofiev, 2019).

The literature review presented in this section show there are multiple pollen source maps produced for different pollen types, regions and using different approaches and data. Despite this, they are rarely applied to, and intercompared using atmospheric

dispersion models. This study aims to evaluate three pollen source maps for the birch taxon using an atmospheric dispersion model and study the effect on the model results by combining these source maps with pollen data. The maps and model are described in the methods section below.

## 2    Methods

### 2.1    Birch pollen source maps

Three birch pollen source maps have been chosen in this study: (1) a map derived by combining land cover data and forest inventory; (2) a map obtained from land cover data and calibrated using model simulations and pollen observations; (3) a statistical map resulting from analysis of forest inventory and forest plot data. The maps have been pre-processed for two mod- elling domains: P15 and T15. P15 covers part of Spain in the south to part of Scandinavia in the North, with Denmark near the center of the model domain (Fig. 1a). T15 covers Finland, parts of Sweden, North-West Russia, Baltic States, Belarus (Fig. 1b).

Map 1 (see Fig. 2a,b) has been obtained by combining three datasets: Global Land Cover Characteristics (GLCC, edc2.usgs.gov/glcc) version 2 (Belward et al., 1999), European Forest Institute tree cover (EFI, Päivinen et al. (2001)) and Tree Species Inventory (TSI, Skjøth et al. (2008)). GLCC has global coverage and includes geographical distribution of land cover, soil type and other various properties with 1 km horizontal resolution. EFI covers Europe, has 1 km horizontal resolution and it contains spatial





distribution of forest with sub-classes, i.e. broadleaved forest and coniferous forest. TSI is presented by 39 types of tree species (birch, oak, alder, etc.) and covers Europe with 50 km horizontal resolution and it is based on national forest inventories and national statistics. The following layers have been extracted from these databases and combined to derive the birch pollen source: land-mask, forest, vegetation, urban territories (from GLCC), broadleaved forest (from EFI) and birch forest fraction

in broadleaved forest (from TSI). For details of the combining procedure see (Kurganskiy et al., 2015; Kurganskiy, 2017).

Map 2 (see Fig. 2c,d) is a product of two datasets and a calibration using model simulations. The two datasets are EFI tree cover and ECOCLIMAP (Masson et al., 2003). ECOCLIMAP is a global dataset with 1 km resolution and it includes 215 ecosystems. The model simulations have been performed with the System for Integrated modeLing of Atmospheric coMposition (SILAM, http://silam.fmi.fi, Sofiev et al. (2015b)) and pollen observation data for several years to ensure average unbiased

representation of the Seasonal Pollen Integral (SPIn) with the SILAM model, where SPIn is the integral of pollen concentrations over time (Galán et al., 2017). The calibrated map is calculated for a 1 km grid covering the whole Europe with nearby regions. Details of the map calibration procedure is described by Prank et al. (2013) using ragweed and its application in the SILAM model on birch can be found in Sofiev (2017).

Map 3 (see Fig. 2e,f) is the result of combining ICP-Level-I forest plot data (www.icp-forests.org) and National Forest Inven-

tory (NFI) plot data as well as the NFI statistics (Brus et al., 2012). Two techniques were used to derive the map: compositional kriging in areas with NFI plot data and regression modelling outside of the areas. The National Forest Inventory statistics was used to scale the regression model results. The map has 1 km horizontal resolution and covers the most part of Europe. However, information about birch pollen sources in Russia is missing in Map 3. Therefore, the Russian sources were extracted from Map 1 and combined with Map 3 to fill the gap.

The maps were introduced to the Enviro-HIRLAM (Environment – High Resolution Limited Area Model) as input data to simulate birch pollen concentrations for the modelling domains.

## 2.2 Model description

Enviro-HIRLAM is an online-coupled meteorology-chemistry model allowing to take into account spatial-temporal evolution of atmospheric chemical (Korsholm, 2009; Baklanov et al., 2017) and biological (Kurganskiy, 2017) aerosols simulated in-

side the meteorological HIRLAM model (Undén et al., 2002). The current version of the Enviro-HIRLAM model is based on HIRLAM v7.2 (https://hirlam.org). The detailed model description of all components can be found in (Korsholm, 2009; Kurganskiy, 2017; Baklanov et al., 2017) while this article is focused on description of the Enviro-HIRLAM features relevant to pollen. In order to simulate birch pollen concentrations Enviro-HIRLAM requires external information/data on the spatial pollen source distribution, phenology, i.e. start of birch flowering, pollen production, characteristics of pollen release

(emissions) within the given study region, where the spatial pollen source distribution is represented by the maps described above in section 2.1. Atmospheric transport and dispersion as well as dry and wet depositions are calculated within Enviro-HIRLAM at each time step. In numerical models, the start of birch pollen flowering is often determined by applying simple crop growth models (e.g. McMaster and Wilhelm (1997)), here the Growing Degree-Day (GDD) method is implemented in Enviro-HIRLAM. GDD is based on accumulation of 2-meter daily mean air temperatures above a given threshold (cut-off





temperature) as this is the WMO definition of GDD. It is assumed here that birch flowering starts as soon as the accumulated temperature (GDD) reaches a certain value depending on geographical location. These temperature sum threshold maps have been produced by (Sofiev et al., 2013) and have been implemented and used as input data for the GDD parameterization in Enviro-HIRLAM. The data were obtained by the fitting procedure applied using the observed leaf unfolding dates (Siljamo

et al., 2008) and modelled ones by the GDD parameterization and ECMWF's ERA-40 reanalysis air temperature data (Uppala et al., 2005) utilized as input for GDD. It was shown in Kurganskiy (2017) that the temperature sum threshold maps are applicable for use to simulate of the birch flowering start in Enviro-HIRLAM, even though another model (ERA-40) has been originally used to obtain the maps. Birch pollen emission is meteorology-dependent and uses parameterizations proposed by (Sofiev et al., 2013). It is based on dimensionless functions correcting the seasonal pollen productivity, here chosen to be

$3.7 \times 10^8$ pollen m$^{-2}$ season$^{-1}$) as a default value obtained for SILAM. The main meteorological parameters affecting the emission are air temperature, relative humidity, wind speed and accumulated precipitation. The emitted birch pollen particles are subject to atmospheric transport which is parameterized in the same way as for aerosols using the Locally Mass Conserving Semi- Lagrangian (LMCSL) scheme (Kaas, 2008; Sørensen et al., 2013). The scheme provides proper mass conservation, shape preservation, and multi-tracer efficiency required from numerical point of view. Dry deposition of birch pollen particles

is determined by gravitational settling (Seinfeld and Pandis, 2006). The gravitational settling parameterization is based on calculation of settling velocity according to Stokes law and taking into account density of birch pollen particles (800 kg m$^{-3}$) and an estimated size of 22 µm (e.g., Mäkelä, 1996). Wet deposition distinguishes between in-cloud (Stier et al., 2005) and below-cloud (Baklanov and Sørensen, 2001) scavenging, i.e. removal of pollen particles from the atmosphere by precipitation. Both scavenging types are parameterized through the scavenging coefficients used for aerosol particles with r $\geq$ 10 µm.

**2.3  Model configuration**

The Enviro-HIRLAM model is used to calculate pollen concentrations in two different modelling domains shown in Fig. 1. Both domains have about 15 km horizontal resolution and 40 hybrid vertical levels. Input meteorological initial and boundary conditions were taken from the ECMWF-IFS model (Persson, 2011) with 15 km resolution and 6-hour interval. The birch pollen emission module has the same settings as described in (Sofiev et al., 2013; Siljamo, 2013). The Enviro-HIRLAM model

has been run for the birch pollen season 2006. The two model domains have been simulated with three types of calculations: 1) A standard calculation, 2) A calculation that includes a correction for 2-meter air temperature ($T_{2m}$) bias between assimilated and simulated $T_{2m}$ (termed COR), 3) A calculation that includes both the correction for temperature and a grid based scaling factor for pollen emissions (termed SCF). These calculations were performed using each of the selected pollen source maps, hence 9 model calculations for each of the two domains. The model results from model simulations 2 and 3 are presented in

the results section. The scaling factor is based on the ratio between simulated and observed Seasonal Pollen Integral (SPIn) (see next section) from the simulations of type 2. This provides 15 point values that have been interpolated to the entire model domain, thereby providing a grid based scaling factor for each of the tree source maps. The grid based scaling factor has then been extracted for each model domain, thus providing 6 grid based scaling factors (Fig. 3). The principal difference between the runs is that the COR ones include a constant value for birch pollen productivity ($3.7 \times 10^8$ pollen m$^{-2}$ season$^{-1}$)) in each





grid cell whereas the productivity is calibrated using the grid based scaling factor (shown in Fig. 3) in the SCF runs. The model results were compared with the pollen observation data at 12 measurement sites located in Finland, Denmark and Russia (see Fig. 1) for all model runs.

## 2.4  Used observations and statistical evaluation of model results

Daily concentrations of birch pollen were used from 12 sites (Fig. 1a,b) and furthermore annual SPIn were obtained from 3 additional sites in Lithuania, published by Veriankaite (2010) in order to strengthen the calculation of the grid based scaling factor (section 2.3). The observational data were obtained using 7-day volumetric samplers (HIRST, 1952) and analysed according to standard methods in aerobiology (Galán et al., 2014). Station names and coordinates are presented in the appendix section along with statistical summaries for each stations (see Tables A1–A4), while overall results are presented in the results section.

The statistical evaluation includes standard and threshold-based statistical approaches, e.g. previously used by (Siljamo et al., 2013; Sofiev et al., 2015a) and calculated in this paper for each map (1,2,3) and run (COR, SCF) types. The start and end of the birch pollen season are calculated using the retrospective method according to Nilsson and Persson (1981), here defined as the dates when the total sums of birch pollen concentrations (SPIn) reach 5 % and 95 %, correspondingly. The criteria are chosen to provide filtration of LRT episodes which have significant impact in Northern Europe, especially in the beginning of the birch

pollen season (Ranta et al., 2006; Skjøth et al., 2007; Mahura et al., 2007; Veriankaite, 2010). The standard statistical approach includes calculations of correlation coefficient (R), coefficient of determination ($R^2$), mean bias (MB) and root mean squared error (RMSE). The $R^2$ values are shown on the time series plots for each station, map and run type as well as on the global scatter plots. The standard statistical metrics have been calculated for the "main" pollen season (30 April - 15 June, 2006) due to non-ergodicity (i.e. non-similarity in space and time) and non-stationarity of the birch pollen time series (Sofiev et al.,

2015a). The threshold-based statistical approach (Siljamo et al., 2013) comprises calculations of the model accuracy (MA), probability of detection (POD), false alarm ratio (FAR), probability of false detection (POFD) and odds ratio (OR) using the threshold value for birch pollen concentrations $C_{th} = 50$ pollen m$^{-3}$. The threshold value $C_{th}$ is chosen since most of the pollen allergy sensitive population might start suffering from allergic reactions when daily mean birch pollen concentration, $C \geq C_{th}$ in the air (Jantunen et al., 2012). Additionally, the modelled and observed birch pollen concentrations have been sorted into

classes in order to estimate the number of cases for low (1–10 pollen m$^{-3}$), moderate (10–100 pollen m$^{-3}$), high (100–1000 pollen m$^{-3}$), and very high (> 1000 pollen m$^{-3}$) birch pollen concentrations. The sorted data plotted with corresponding standard errors are shown in the result section. Hit rates (HR) are calculated for the classes and their results are shown in tables in the results section. The results of statistical significance test (chi-squared test) performed for the HR values are also discussed.

## 3  Results

The simulated time series of birch pollen concentrations have been compared against the selected pollen observation sites by determining the start and end of the birch pollen season and applying the standard statistics as well as threshold-based statistics described in the methods section. The modelled and observed time series are also shown in Fig. 4–5 for COR and SCF runs,





respectively. The start of the birch pollen season is simulated quite well for both runs and all maps. Station-wise the bias does not exceed 2–3 days at all stations and indicates 2 days too early flowering on average for all maps. Simulation of the last flowering day turns to be the most challenging for Enviro-HIRLAM. The bias shows too late flowering at almost all stations. On average, it varies from 15 days (Map 3) to 9 days (Map 2) according to the COR run. Rescaling of the pollen productivity

(SCF run) reduced the bias by 2-3 days for each map. The largest biases for the last flowering day are found at five Finnish sites (Helsinki, Joutseno, Kangasala, Vaasa and Kuopio) and two Russian sites (Moscow and Smolensk) and they are related to overestimation of the observed birch pollen concentrations at the end of the season for both COR and SCF runs (see Fig. 4 and 5). It can be addressed to large variations in the year to year pollen production in that region which, in turn, introduce a large bias in the model simulations.

Generally, the time series of the modelled birch pollen concentrations ( Fig. 4 and 5) show good agreement with observations for all maps at most of the selected stations. COR run indicates that the model behaves similar with respect to describing variance in observations (see $R^2$ values in Fig. 4), except some stations (e.g. Moscow, Helsinki) where Map 2 provides higher $R^2$ values in comparison with Map 1 and Map 3. The SCF run did not affect the $R^2$ values significantly, but it reduced the mean bias (MB) at the stations. The highest $R^2$ values are found at two Danish sites (Copenhagen and Viborg) where the model

explains more than 70 % of variance found in observations for all considered maps and model runs. For more detailed statistical summaries at the stations the reader is referred to tables in the appendix section.

The global standard statistical metrics for all stations are presented in Table 1. According to the COR run the model performs slightly better with Map 2. It is supported by higher correlation coefficient (R = 0.59) and lower mean bias (MB = 69.08 pollen m$^{-3}$) in comparison with two other maps. One can assume that this result should be expected since pollen observa-

tions are already included to the methodology used to obtain Map 2. Rescaling of the pollen productivity (SCF run) provides improvement of the Enviro-HIRLAM model performance by increasing the R values (R = 0.72 for Map 1,2 and R = 0.68 for Map 3) and decreasing the MB values. However, RMSE is large (i.e. > 1000 pollen m$^{-3}$) for all maps and both model runs. It can be explained by sensitivity of RMSE to large differences between modelled and observed values. The large bias for the last flowering day simulation contributes significantly to the RMSE values.

The global scatter plots (Fig. 6) also indicate improvements in the modelled-observed concentration agreement expressed in increase of the $R^2$ for the SCF model setup. As seen the coefficient of determination do not exceed 0.34 (Map 2) for all maps in COR run. SCF run provides $R^2$ values around 0.5 for all maps. This means that the model can explain about 50 % of variance found in observations after the rescaling using any of the maps.

The threshold-based statistical metrics calculated for each map and model run are shown in Table 2. The results indicate simi-

lar behavior of the model when comparing performance of the maps. The values of the statistical metrics are slightly different between the maps, but the difference does not exceed 7 %. The model accuracy (MA) – the fraction of correct simulations – is higher than 70 % for all model simulations. The POD values show that the Enviro-HIRLAM model simulates the high concentrations (> 50 pollen m$^{-3}$) correctly in more than 70 % of cases. The fraction of incorrect simulations for the high concentrations (FAR values) are less than 30 % for the model setups. The POFD values show that the fraction of the cases when

concentrations are simulated as high, but observed as low (i.e. < 50 pollen m$^{-3}$) is more than 30 % for the COR runs and it





decreases below 30 % after rescaling of the pollen productivity (SCF run). As expected the odds ratio (OD) values increases in SCF run for all maps. The modelled and observed birch pollen concentrations have been grouped in order to estimate the number of cases for low, moderate, high and very high birch pollen concentrations (see Fig. 7). As seen the Enviro-HIRLAM model simulates the low and very high pollen concentrations quite well with the difference less than ± 5 % between the modelled

and observed number of cases for both COR (Fig. 7a) and SCF (Fig. 7b) runs and all maps. The largest differences between the modelled and observed number of cases are found for the moderate and high pollen concentrations. Map 2 underestimates the number of cases for the moderate concentrations by 10 % whereas Map 3 overestimates the high ones by more than 10 % (COR run). SCF run slightly improves the results for the moderate and high number of cases.

The hit rate (HR) calculations (see Tables 3–4) show that simulations of the moderate and high birch pollen concentrations is

the most challenging for all model setups. SCF run (Table 4) indicates improvements of the model score for the moderate and high birch pollen concentrations. However, the hit rates still do not exceed 50 % for all maps. This feature is also visible on the scatter plots in Fig. 6. The HR values for the low pollen concentrations are quite high and exceed 90 % for COR and 80 % for SCF runs. The very high pollen concentrations are simulated slightly better by using Map 3 with HR = 70 %. However, according to the chi square test the differences between the HR values for different maps are not statistically significant for

both COR and SCF runs.

## 4   Discussion

Analysis of the start of birch pollen season showed good model performance for all maps with the bias not exceeding 2 days on average. Such error is acceptable by state-of-the-art atmospheric dispersion models (e.g., Sofiev et al., 2015a). The analysis did not reveal significant dependency of the start of the birch pollen season on the underlying pollen source map and/or pollen

productivity. This could be explained by the fact that the start of the birch pollen season is calculated using meteorological parameters (i.e. temperate sum thresholds) employed in GDD parameterization in the model. However, long distance transport of birch pollen has often been observed in that region (e.g., Hjelmroos, 1992; Mahura et al., 2007) which can systematically affect the start of the pollen season (Skjøth et al., 2007). It could therefore be assumed that changing the source term in remote regions such as Poland could affect the start of the calculated pollen season in Denmark and Sweden. This phenomenon

was not observed, which may be explained by the fact that the scaling factor for Poland was near to unity for all three maps, while the scaling factor for Copenhagen and the near surroundings was substantially higher. This observation therefore complements the suggestion from other studies that long distance transport of pollen is an episodic phenomenon (e.g., Smith et al., 2008; Fernández-Rodríguez et al., 2014) and that the overall contribution to locally observed birch pollen concentration is due to sources found within the region (Sofiev, 2017). The last flowering day simulation was the most challenging for Enviro-

HIRLAM. Map 2 showed the lowest bias estimated as 6 days (too late flowering end) on average for the stations according to the SCF runs. However, such bias is a known feature for state-of-the-art atmospheric models (e.g., Sofiev et al., 2015a). Station-wise the Finish and Russian sites had the bias ≥ 2 weeks in all model runs, which complements a multi-model ensemble, that demonstrated generally good results for Europe but also highlighted that large variations in the year to year pollen





production in that region can introduce a large bias in the model simulations (Sofiev et al., 2015a). Large annual variations in the birch pollen integral is well known for a number of European countries (Latałowa et al., 2002; Dahl et al., 2013; Piotrowska and Kaszewski, 2011), as these annual fluctuations are generally not accounted for in the atmospheric models (e.g., Ritenberga et al., 2018).

The standard statistical metrics indicated a slightly better performance of the Enviro-HIRLAM with Map 2 on average for the stations according to the COR run. However, even for this map, the model explained less than 35 % of variance in observations. It is therefore plausible to attribute the remaining part of the disagreement to small-scale patterns of the birch habitation, which are neither captured by existing calibration methods nor satisfactorily reproduced by the land-cover datasets. In this study, a scaling of the pollen productivity improved the model results significantly for all source terms and provided near to identical

results for Map 1 and Map 2. This suggest, that accurate exposure calculations that use dispersion models preferably should use data fusion that combine a detailed inventory based source term (e.g., Skjøth et al., 2008; Pekkarinen et al., 2009; Sofiev, 2017) along with observed pollen concentration covering the period of interest.

     The threshold-based statistics calculated relatively to one threshold value ($C_{th}$ = 50 pollen m$^{-3}$) showed a good model agreement with observations independently of the map used. The threshold-based analysis was also done for low (1–10 pollen m$^{-3}$),

moderate (10–100 pollen m$^{-3}$), high (100–1000 pollen m$^{-3}$) and very high (> 1000 pollen m$^{-3}$) birch pollen concentrations. According to the analysis simulations of the moderate and high birch pollen concentrations appeared to be the most challenging for Enviro-HIRLAM for all maps. The hit rates were less than 50 % for all model setups. It reveals a need in improvement of the rescaling of birch pollen productivity by introducing more observational points and/or performing more iterations. The differences of the HR values appeared to be not significant during intercomparsion of the maps. This study concerning birch

pollen complements other studies on ragweed (Prank et al., 2013; Hamaoui-Laguel et al., 2015), that also demonstrated the need for recalibration of the source term. However, it has since been shown by Zink et al. (2017) that source terms combining pollen data from several years with detailed land cover data can outperform other approaches minimizing the need local calibrations. This approach is also limited by the fact that only few regions have a sufficiently high number of pollen observations (Zink et al., 2017). Add to this that abrupt annual changes in the underlying pollen integral can be caused by changes in the

pollen emission (Bonini et al., 2015, 2018).

     The insufficient density of the observational network also became the methodological limitation of the current study: we have used the same data from the same stations and for the same years for both calibration and evaluation. However, we separated them frequency-wise: calibration uses annual or multi-annual average whereas the evaluation primarily concerned correlation and RMSE.

## 5   Conclusions

The aim of the study was to evaluate three birch pollen source maps using the atmospheric dispersion model Enviro-HIRLAM. The evaluation has been performed for 12 pollen observation sites located in Denmark, Finland and Russia. The modelled

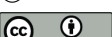



and observed time series of birch pollen concentrations have been analyzed for the start and end of flowering as well as by calculating the standard and threshold-based statistical metrics for two sets of the Enviro-HIRLAM model runs: without and with calibration using pollen observations.

The analysis did not reveal significant dependency of the start/end of the birch pollen season on the underlying pollen source
map. The statistical analysis showed a good model agreement with the observed birch pollen concentrations in the region studied. The model can explain up to 50 % of variance found in observations after the calibration with all maps. It was shown that the remaining part of the disagreement should be addressed to small-scale patterns of the birch habitation, which are neither captured by existing calibration methods nor satisfactorily reproduced by the land-cover datasets. The analysis also revealed that the insufficient density of the observational network was the methodological limitation of the study.

Generally, it was shown that calibration of the maps using pollen observations covering the investigation year significantly improved the model performance for all three maps. The findings also indicate the large sensitivity of the model results to the source maps and agree well with other studies on birch showing that pollen or hybrid-based source maps provide the best model performance. This study highlights the importance of including pollen data in the production of source maps for pollen dispersion modelling and for exposure studies.

*Code and data availability.* The model code and data are available upon request from the authors.

*Author contributions.* AK performed the model simulations and analysis, created graphical outputs and the initial paper draft; CAS participated in the analysis, preparation of the initial paper draft and contributed to writing the paper; EK, AB and SS supervised AK, provided research advices and infrastructure and contributed to the paper writing, MS participated in the experiment planning and execution and contributed to the paper writing; AS, ES provided the pollen observation data and contributed to the paper writing.

*Competing interests.* The authors declare that they have no conflict of interest.

*Acknowledgements.* The authors are greatly thankful to Dr. Suleiman Mostamandy (RSHU/KAUST), Dr. Brain Højen-Sørensen (NBI UC/FCOO) and Dr. Roman Nuterman (NBI UC) for useful advices on Enviro-HIRLAM code development; Dr. Alexander Mahura (DMI/UH) and Alix Rasmussen (DMI) for fruitful discussions of the birch pollen modelling issues; European Forest Institute (EFI) - for providing broadleaved forest data; Adomas Mazeikis - for useful advices on GIS analysis; Danish Asthma Allergy Association - for birch pollen ob-
servation data; RSHU - for providing HPC facilities. The research leading to the results presented in this paper has received funding from Scholarship of the President of the Russian Federation for Students and PhD Students Training Abroad, EuMetChem - COST Action ES1004 as well as NBI UC, RSHU internal grants. SILAM pollen module was developed with Finnish Academy projects APTA (N 266314) and PS4A (No 318194). Support of Copernicus CAMS-50 service is kindly acknowledged.



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

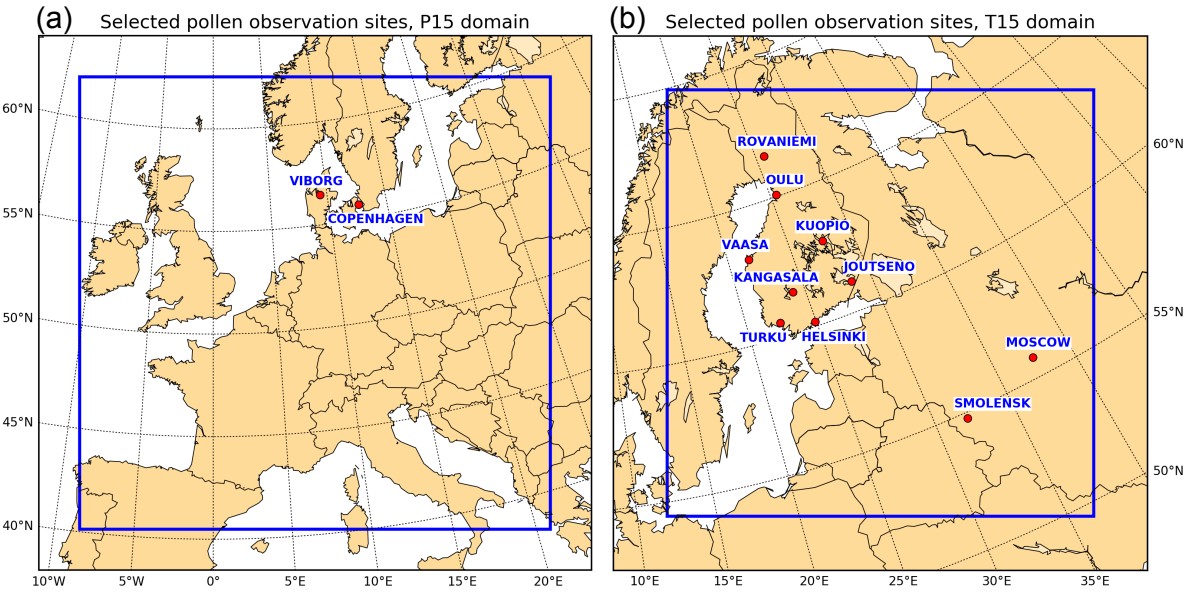

**Figure 1.** Location of the selected birch pollen observation sites with boundaries of the modelling domains: P15 (a) and T15 (b).

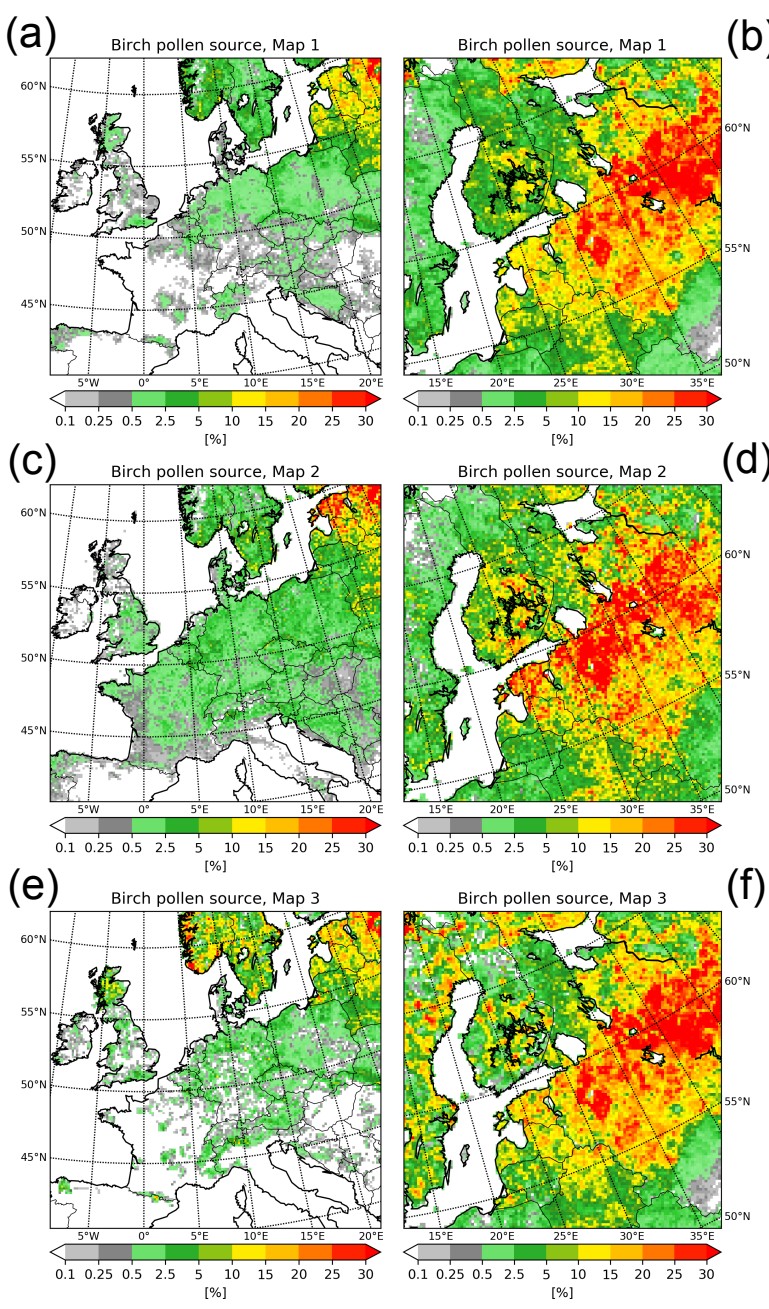

**Figure 2.** Birch pollen source maps processed for two different modelling domains P15 (left) and T15 (right) with 15 km grid resolution. The upper panel corresponds to Map 1 (a,b), the middle panel - to Map 2 (c,d) and the lower panel - to Map 3 (e,f). The maps are shown in [%].



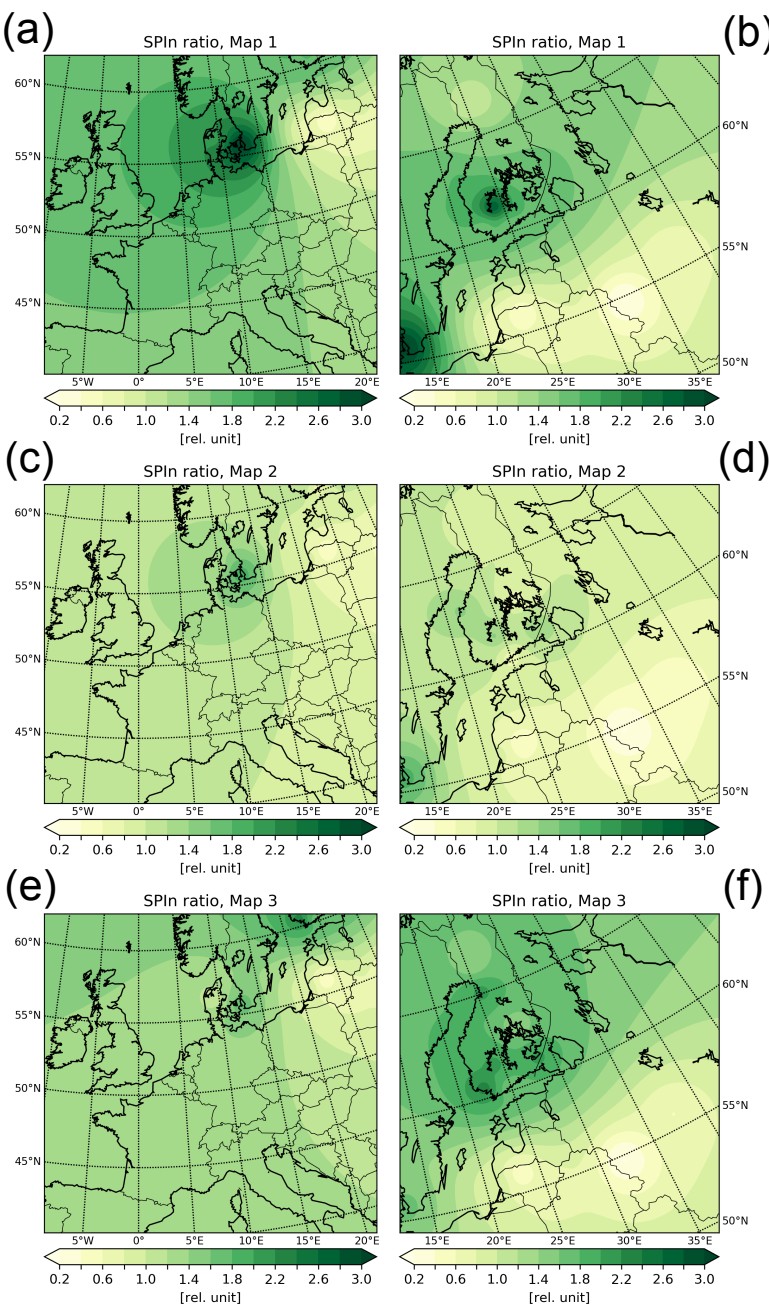

**Figure 3.** SPIn ratio used to calibrate pollen productivity in the SCF runs for two different domains P15 (left) and T15 (right). The upper panel corresponds to Map 1 (a,b), the middle panel - to Map 2 (c,d) and the lower panel - to Map 3 (e,f).





**Figure 4.** Time series of the modelled and observed birch pollen daily mean concentrations for the selected pollen observation sites. The modelled concentrations correspond to each simulated map according to the COR run. A base-10 log scale is used for the Y axis.



**Figure 5.** Time series of the modelled and observed birch pollen daily mean concentrations for the selected pollen observation sites. The modelled concentrations correspond to each simulated map according to the SCF run. A base-10 log scale is used for the Y axis.



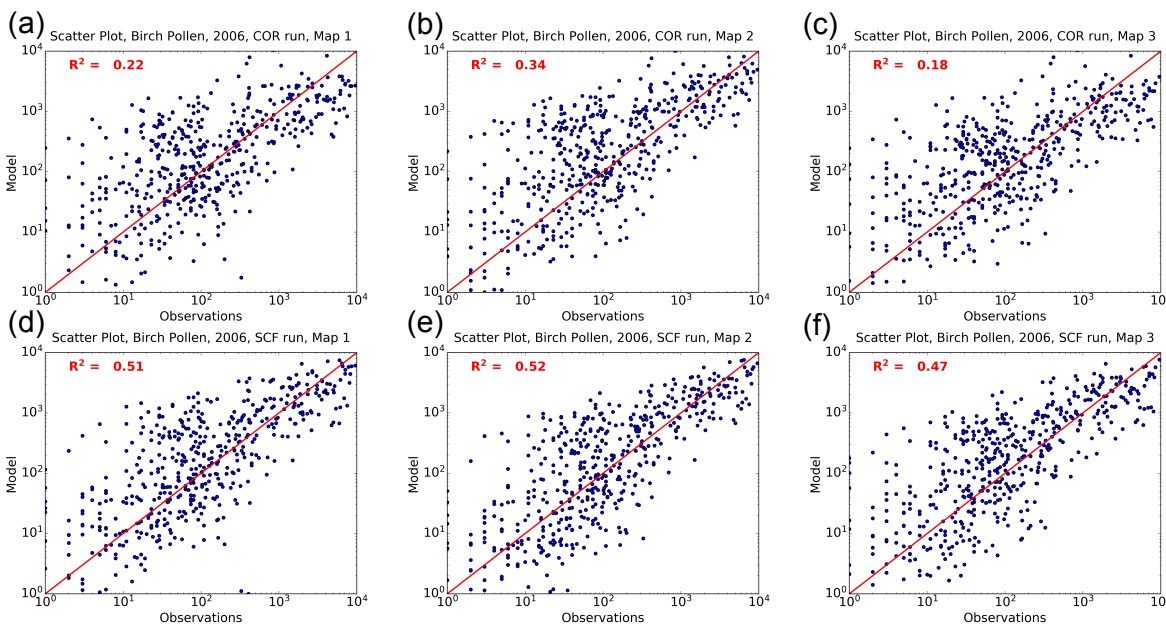

**Figure 6.** Scatter plots for birch pollen concentrations according to comparing COR (the upper panel) and SCF (the lower panel) model runs with observations for Map 1 (a,d), Map 2 (b,e) and Map 3 (c,f).

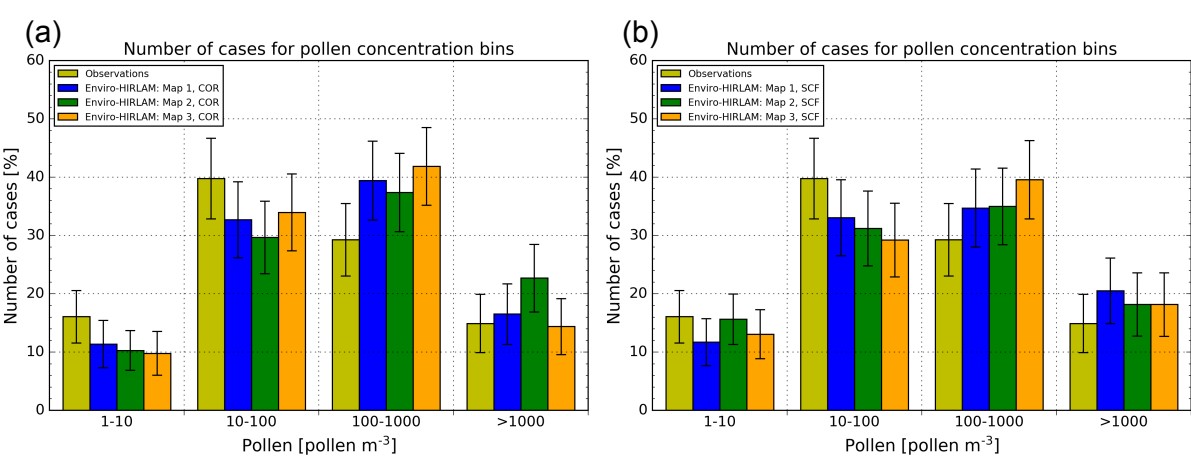

**Figure 7.** Histogram of the observed and modelled birch pollen concentrations sorted into low (1–10 pollen m$^{-3}$), moderate (10–100) pollen m$^{-3}$, high (100–1000 pollen m$^{-3}$) and very high (> 1000 pollen m$^{-3}$) classes. The left panel (a) corresponds to COR and the rght panel (b) - to the SCF model simulations.





**Table 1.** Global "standard" statistical metrics
obtained by comparing the modelled ($\overline{C_m}$) and observed ($\overline{C_o}$) daily mean concentrations.
$\overline{C_m}$, $\overline{C_o}$, MB and RMSE are given in [pollen m$^{-3}$], p-value $< 0.01$

| Run | R | $\overline{C_m}$ | $\overline{C_o}$ | MB | RMSE |
|---|---|---|---|---|---|
| COR, Map 1 | 0.47 | 545.89 | 683.63 | -137.74 | 1587.81 |
| COR, Map 2 | 0.59 | 752.71 | 683.63 | 69.08 | 1465.79 |
| COR, Map 3 | 0.42 | 509.49 | 683.63 | -174.14 | 1639.47 |
| SCF, Map 1 | 0.72 | 661.64 | 683.63 | -21.99 | 1229.16 |
| SCF, Map 2 | 0.72 | 630.02 | 683.63 | -53.61 | 1223.95 |
| SCF, Map 3 | 0.68 | 614.41 | 683.63 | -69.22 | 1290.04 |



**Table 2.** The threshold-based statistical metrics obtained by comparing model simulations with observed daily mean concentrations. MA, POD, FAR and POFD are given in [%].

| Run, map | MA | POD | FAR | POFD | OR |
|---|---|---|---|---|---|
| COR, Map 1 | 76.3 | 74.6 | 25.4 | 31.9 | 2.3 |
| COR, Map 2 | 74.7 | 71.1 | 28.9 | 38.3 | 1.9 |
| COR, Map 3 | 75.4 | 72.7 | 27.3 | 35.0 | 2.1 |
| SCF, Map 1 | 79.5 | 77.7 | 22.3 | 27.6 | 2.8 |
| SCF, Map 2 | 79.8 | 79.1 | 20.9 | 24.8 | 3.2 |
| SCF, Map 3 | 79.7 | 77.1 | 22.9 | 29.1 | 2.7 |





**Table 3.** Hit rates of the Enviro-HIRLAM birch pollen simulations (COR run) for classes:

zero or low, moderate, high and very high.

The classes are based on daily mean pollen concentrations and presented in units: [pollen $m^{-3}$].

| Observation Model | 0–10 | 10–100 | 100–1000 | > 1000 |
|---|---|---|---|---|
| COR run, Map 1 | | | | |
| 0–10 | 94.5 % | 4.6 % | 0.9 % | 0.0 % |
| 10–100 | 29.1 % | 46.0 % | 24.9 % | 0.0 % |
| 100–1000 | 6.1 % | 41.3 % | 38.5 % | 14.1 % |
| > 1000 | 0.0 % | 11.5 % | 25.3 % | 63.2 % |
| COR run, Map 2 | | | | |
| 0–10 | 95.2 % | 3.9 % | 0.9 % | 0.0 % |
| 10–100 | 29.0 % | 47.9 % | 23.1 % | 0.0 % |
| 100–1000 | 9.2 % | 44.4 % | 37.7 % | 8.7 % |
| > 1000 | 0.0 % | 15.1 % | 28.6 % | 56.3 % |
| COR run, Map 3 | | | | |
| 0–10 | 95.0 % | 4.2 % | 0.8 % | 0.0 % |
| 10–100 | 34.0 % | 44.2 % | 21.8 % | 0.0 % |
| 100–1000 | 6.9 % | 40.5 % | 40.1 % | 12.5 % |
| > 1000 | 0.0 % | 10.0 % | 20.0 % | 70.0 % |



**Table 4.** Hit rates of the Enviro-HIRLAM birch pollen simulations (SCF run) for classes:

zero or low, moderate, high and very high.

The classes are based on daily mean pollen concentrations and presented in units: [pollen $\mathrm{m}^{-3}$].

| Observation Model | 0–10 | 10–100 | 100–1000 | > 1000 |
|---|---|---|---|---|
| SCF run, Map 1 | | | | |
| 0–10 | 85.5 % | 12.1 % | 2.4 % | 0.0 % |
| 10–100 | 32.0 % | 49.5 % | 18.5 % | 0.0 % |
| 100–1000 | 5.3 % | 40.0 % | 46.8 % | 7.9 % |
| > 1000 | 0.0 % | 12.7 % | 23.6 % | 63.7 % |
| SCF run, Map 2 | | | | |
| 0–10 | 84.8 % | 13.2 % | 2.0 % | 0.0 % |
| 10–100 | 29.6 % | 48.5 % | 21.9 % | 0.0 % |
| 100–1000 | 5.7 % | 41.8 % | 46.3 % | 6.2 % |
| > 1000 | 0.0 % | 11.4 % | 23.7 % | 64.9 % |
| SCF run, Map 3 | | | | |
| 0–10 | 84.0 % | 14.3 % | 1.7 % | 0.0 % |
| 10–100 | 34.9 % | 46.3 % | 18.8 % | 0.0 % |
| 100–1000 | 6.4 % | 42.7 % | 44.0 % | 6.9 % |
| > 1000 | 0.0 % | 5.1 % | 24.2 % | 70.7 % |



# Appendix A

**Table A1.** Birch pollen statistical metrics for the selected observation sites: COR run.
$\overline{C_m}$, $\overline{C_o}$, MB, RMSE are in pollen $m^{-3}$; p-value $< 0.05$.

| St. name | Map | Lat | Lon | R | $\overline{C_m}$ | $\overline{C_o}$ | MB | RMSE |
|---|---|---|---|---|---|---|---|---|
| Helsinki | | 60.17° N | 24.90° E | | | | | |
| | Map 1 | | | 0.80 | 574.69 | 879.28 | -299.59 | 1269.56 |
| | Map 2 | | | 0.89 | 1233.00 | 879.28 | 353.72 | 895.53 |
| | Map 3 | | | 0.70 | 515.97 | 879.28 | -363.31 | 1412.44 |
| Joutseno | | 61.10° N | 28.50° E | | | | | |
| | Map 1 | | | 0.57 | 838.61 | 1345.81 | -507.2 | 2865.20 |
| | Map 2 | | | 0.46 | 1002.84 | 1345.81 | -342.97 | 2945.21 |
| | Map 3 | | | 0.45 | 659.18 | 1345.81 | -686.63 | 3070.78 |
| Kangasala | | 61.47° N | 24.08° E | | | | | |
| | Map 1 | | | 0.86 | 569.86 | 1442.55 | -872.69 | 2369.17 |
| | Map 2 | | | 0.87 | 1018.23 | 1442.55 | -424.32 | 1841.15 |
| | Map 3 | | | 0.84 | 701.75 | 1442.55 | -740.8 | 2252.54 |
| Kuopio | | 62.88° N | 27.63° E | | | | | |
| | Map 1 | | | 0.51 | 672.75 | 777.23 | -104.48 | 1627.23 |
| | Map 2 | | | 0.60 | 1032.18 | 777.23 | 254.95 | 1508.64 |
| | Map 3 | | | 0.49 | 530.67 | 777.23 | -246.56 | 1665.51 |
| Oulu | | 65.07° N | 25.52° E | | | | | |
| | Map 1 | | | 0.63 | 469.77 | 682.43 | -212.66 | 845.99 |
| | Map 2 | | | 0.64 | 686.56 | 682.43 | 4.13 | 883.44 |
| | Map 3 | | | 0.55 | 333.85 | 682.43 | -348.58 | 950.66 |
| Rovaniemi | | 66.55° N | 25.73° E | | | | | |
| | Map 1 | | | 0.42 | 302.92 | 309.64 | -6.72 | 574.93 |
| | Map 2 | | | 0.35 | 324.04 | 309.64 | 14.40 | 646.57 |
| | Map 3 | | | 0.36 | 211.50 | 309.64 | -98.14 | 544.39 |





**Table A2.** Birch pollen statistical metrics for the selected observation sites: COR run.
$\overline{C_m}$, $\overline{C_o}$, MB, RMSE are in pollen m$^{-3}$; p-value $< 0.05$.

| St. name | Map | Lat | Lon | R | $\overline{C_m}$ | $\overline{C_o}$ | MB | RMSE |
|---|---|---|---|---|---|---|---|---|
| Turku | | 60.53° N | 22.47° E | | | | | |
| | Map 1 | | | 0.87 | 548.80 | 925.02 | -376.22 | 1269.02 |
| | Map 2 | | | 0.86 | 862.73 | 925.02 | -62.29 | 1011.47 |
| | Map 3 | | | 0.76 | 404.71 | 925.02 | -520.31 | 1605.97 |
| Vaasa | | 63.10° N | 21.62° E | | | | | |
| | Map 1 | | | 0.56 | 307.10 | 548.43 | -241.33 | 902.99 |
| | Map 2 | | | 0.56 | 389.07 | 548.43 | -159.36 | 871.11 |
| | Map 3 | | | 0.57 | 294.52 | 548.43 | -253.91 | 913.76 |
| Moscow | | 55.74° N | 37.55° E | | | | | |
| | Map 1 | | | 0.37 | 647.42 | 388.28 | 259.14 | 1269.87 |
| | Map 2 | | | 0.59 | 733.42 | 388.28 | 345.14 | 1100.18 |
| | Map 3 | | | 0.37 | 647.40 | 388.28 | 259.12 | 1270.04 |
| Smolensk | | 54.78° N | 32.05° E | | | | | |
| | Map 1 | | | 0.78 | 1382.37 | 259.32 | 1123.05 | 2542.82 |
| | Map 2 | | | 0.77 | 1368.29 | 259.32 | 1108.97 | 2466.38 |
| | Map 3 | | | 0.78 | 1377.17 | 259.32 | 1117.85 | 2537.15 |
| Copenhagen | | 55.72° N | 12.57° E | | | | | |
| | Map 1 | | | 0.93 | 144.63 | 445.83 | -301.2 | 717.53 |
| | Map 2 | | | 0.95 | 231.68 | 445.83 | -214.15 | 560.20 |
| | Map 3 | | | 0.92 | 263.55 | 445.83 | -182.28 | 503.85 |
| Viborg | | 56.45° N | 9.40° E | | | | | |
| | Map 1 | | | 0.94 | 91.72 | 199.70 | -107.98 | 325.90 |
| | Map 2 | | | 0.94 | 150.53 | 199.70 | -49.17 | 229.32 |
| | Map 3 | | | 0.88 | 173.58 | 199.70 | -26.12 | 251.93 |





**Table A3.** Birch pollen statistical metrics for the selected observation sites: SCF run.
$\overline{C_m}$, $\overline{C_o}$, MB, RMSE are in pollen $m^{-3}$; p-value $< 0.05$.

| St. name | Map | Lat | Lon | R | $\overline{C_m}$ | $\overline{C_o}$ | MB | RMSE |
|---|---|---|---|---|---|---|---|---|
| Helsinki | | 60.17° N | 24.90° E | | | | | |
| | Map 1 | | | 0.86 | 841.46 | 879.28 | -37.82 | 942.66 |
| | Map 2 | | | 0.89 | 981.76 | 879.28 | 102.48 | 833.56 |
| | Map 3 | | | 0.79 | 785.22 | 879.28 | -94.06 | 1122.77 |
| Joutseno | | 61.10° N | 28.50° E | | | | | |
| | Map 1 | | | 0.62 | 1224.33 | 1345.81 | -121.48 | 2601.02 |
| | Map 2 | | | 0.54 | 989.56 | 1345.81 | -356.25 | 2823.36 |
| | Map 3 | | | 0.53 | 1015.47 | 1345.81 | -330.34 | 2812.42 |
| Kangasala | | 61.47° N | 24.08° E | | | | | |
| | Map 1 | | | 0.89 | 1100.24 | 1442.55 | -342.31 | 1595.21 |
| | Map 2 | | | 0.89 | 1135.93 | 1442.55 | -306.62 | 1575.66 |
| | Map 3 | | | 0.85 | 1200.14 | 1442.55 | -242.41 | 1633.02 |
| Kuopio | | 62.88° N | 27.63° E | | | | | |
| | Map 1 | | | 0.55 | 953.83 | 777.23 | 176.6 | 1559.38 |
| | Map 2 | | | 0.62 | 817.90 | 777.23 | 40.67 | 1474.81 |
| | Map 3 | | | 0.56 | 755.99 | 777.23 | -21.24 | 1543.05 |
| Oulu | | 65.07° N | 25.52° E | | | | | |
| | Map 1 | | | 0.63 | 669.43 | 682.43 | -13.00 | 876.15 |
| | Map 2 | | | 0.65 | 676.29 | 682.43 | -6.14 | 861.12 |
| | Map 3 | | | 0.60 | 583.34 | 682.43 | -99.09 | 857.18 |
| Rovaniemi | | 66.55° N | 25.73° E | | | | | |
| | Map 1 | | | 0.35 | 368.22 | 309.64 | 58.58 | 691.80 |
| | Map 2 | | | 0.34 | 309.19 | 309.64 | -0.45 | 632.14 |
| | Map 3 | | | 0.34 | 332.60 | 309.64 | 22.96 | 611.50 |





**Table A4.** Birch pollen statistical metrics for the selected observation sites: SCF run.
$\overline{C_m}$, $\overline{C_o}$, MB, RMSE are in pollen m$^{-3}$; p-value $< 0.05$.

| St. name | Map | Lat | Lon | R | $\overline{C_m}$ | $\overline{C_o}$ | MB | RMSE |
|---|---|---|---|---|---|---|---|---|
| Turku | | 60.53° N | 22.47° E | | | | | |
| | Map 1 | | | 0.86 | 958.24 | 925.02 | 33.22 | 1032.33 |
| | Map 2 | | | 0.86 | 924.51 | 925.02 | -0.51 | 1033.30 |
| | Map 3 | | | 0.79 | 732.00 | 925.02 | -193.02 | 1241.52 |
| Vaasa | | 63.10° N | 21.62° E | | | | | |
| | Map 1 | | | 0.60 | 507.24 | 548.43 | -41.19 | 831.66 |
| | Map 2 | | | 0.63 | 475.72 | 548.43 | -72.71 | 811.14 |
| | Map 3 | | | 0.61 | 503.97 | 548.43 | -44.46 | 817.61 |
| Moscow | | 55.74° N | 37.55° E | | | | | |
| | Map 1 | | | 0.34 | 392.03 | 388.28 | 0.01 | 1206.26 |
| | Map 2 | | | 0.57 | 334.68 | 388.28 | -0.16 | 1047.91 |
| | Map 3 | | | 0.33 | 375.92 | 388.28 | -0.03 | 1208.18 |
| Smolensk | | 54.78° N | 32.05° E | | | | | |
| | Map 1 | | | 0.61 | 505.21 | 259.32 | 245.89 | 859.77 |
| | Map 2 | | | 0.65 | 415.25 | 259.32 | 155.93 | 693.79 |
| | Map 3 | | | 0.61 | 494.46 | 259.32 | 235.14 | 854.60 |
| Copenhagen | | 55.72° N | 12.57° E | | | | | |
| | Map 1 | | | 0.92 | 261.39 | 445.83 | -184.44 | 494.49 |
| | Map 2 | | | 0.94 | 312.42 | 445.83 | -133.41 | 407.55 |
| | Map 3 | | | 0.91 | 370.93 | 445.83 | -74.90 | 418.39 |
| Viborg | | 56.45° N | 9.40° E | | | | | |
| | Map 1 | | | 0.91 | 158.00 | 199.70 | -41.7 | 239.74 |
| | Map 2 | | | 0.92 | 187.03 | 199.70 | -12.67 | 201.17 |
| | Map 3 | | | 0.85 | 222.83 | 199.70 | 23.13 | 261.80 |