# Peer review of "Incorporation of pollen data in source maps is vital for pollen dispersion models"

_Atmospheric Chemistry and Physics, 2019_

## Referee Comment (RC1) · Anonymous Referee #1 · 7 Aug 2019

It is my impression that the manuscript both falls within the scope, and is scientifically sound. The topic of the development of source maps, and the impact of these on the final model results, is very relevant for further development. However, this paper requires an intimate knowledge of pollen modelling and detailed reading of a number of background papers, to completely comprehend the analysis. The details of pollen dispersion modelling is not within my main area of expertise, and a thorough assessment of the method, is therefore beyond my current knowledge.

Questions / Comments

Do the authors have any comments on why the correlations are better for the stations in the P15 domain calculations? It is interesting that the "moderate" concentrations is generally underestimated and the "high" is generally overestimated by all combinations

of model/maps. Could the authors elaborate on possible explanations for this? What do the authors think is the explanation for the variation in the scaling factor, and the high values near Copenhagen (Discussion, page 8, line 26)? Is this map relevant for other applications, e.g. for selecting areas where further studies are needed to establish why modelled and observed values do not agree? Page 7, line 14-15 / 20-21 – The Danish sites are mentioned as having the highest R2. Could you comment on why the SCF run appears to result in lower R2 than the COR at these sites, opposite of most other sites. Perhaps in the discussion section.

Minor comments and/or typographical corrections

Page 5, line 7 - "... for use to simulate of the birch flowering season". I suggest to delete "of".

Page 5, line 10 – Reference for details on the choice of the seasonal pollen productivity?

Page 5, line 28 – Could you add further description of the pollen emissions scaling factor? It is not clear whether it is this factor that is further explained in line 30 and onward? If so, please apply the acronym here (line 30). Also, in line 30-31, it is unclear whether the scaling factor is based on the COR-run (simulations of type 2), and then following applied for all maps?

Page 5, line 34 - "... the COR ones..." I suggest to change to "COR-runs".

Page 5, line 34 - Two closing-parenthesis, delete one.

Page 6, line 1 – Again a reference to "the grid based scaling factor". I suggest to consistently use the abbreviation, you introduced on page 5, if it is the same "grid based scaling factor".

Page 6, line 14 – Define LRT-abbreviation as Long Range Transport.

Page 7, line 14 – I suggest to add a reference to appendix A for the referred MB results,

already in this line. I am aware that the reference is listed in the final line of this section, but I would prefer it also listed when the results are first mentioned.

Page 8, line 9 – HR has already been defined as Hit rate (page 6, line 28), so there is no need to write it out here.

Page 8, line 30 – "(too late flowering end")". I suggest to rephrase, e.g. "delay of end of flowering".

Page 9, line 22 - " . . . the need local calibrations. . ..". Please add "for": ". . . the need for local calibrations."

Page 9, line 28-29 – This information borders on "method", could the latter part be moved to methods?

Figures

Fig 1 – I suggest to add a scale bar.

Fig 4-5. What is the order of the charts based on? The first and last appears to be the P15 domain, and the rest, the T15 domain. Perhaps move (a) and (l) next to each other, if there are no other reason behind the order.
* * *

---

## Referee Comment (RC2) · Rachel McInnes (Referee) · 12 Aug 2019

General comments: This is a really nice piece of work which adds to a very topical area of study. It is novel and extends scientific progress in this area. I believe it is relevant to the scope of the Atmospheric Chemistry and Physics journal.

The authors demonstrate the importance of including observational pollen data in producing pollen source maps for pollen dispersion modelling, and they outline the health impacts for the population which is why this modelling is so important.

Specific comments: Section 2.1 – while you have provided information on the datasets used and references to explain the methodological differences between map 1, map 2 and map 3 – it is quite difficult to overview the 3 methods at a glance, in order

to compare and contrast. I suggest that a graphical diagram outlining the method / flowchart for each different map might help the reader understand the differences between the three maps being compared.

Section 2.2& 2.3 – what dates were the models run over? This needs clarified please. E.g. in Section 2.2 when discussion the GDD data - can you clarify what dates were used for the input data? In 2.3 you state: "Input meteorological initial and boundary conditions were taken from the ECMWF-IFS model (Persson, 2011) with 15 km resolution and 6-hour interval" – this would be a good point to say what years the models were run over. When you mention the Birch 2006 season – is that this work, or are you referring to previous work?

Section 2.3 – what was the reason for excluding the method 1 results from the results section?

Section 3 – "The largest differences between the modelled and observed number of cases are found for the moderate and high pollen concentrations." – but are these all within the error bars, see Fig 7?

Section 4 – can you comment in the manuscript on what work you believe would need to be done to try to improve the modelling of the last flowering day?

Section 4 – "that also demonstrated the need for recalibration of the source term. However, it has since been shown by Zink et al. (2017) that source terms combining pollen data from several years with detailed land cover data can outperform other approaches minimizing the need local calibrations. " - Linked to my earlier question - has this study only been carried out for the one year? Please clarify and comment on the impact of how long a period you are doing this analysis over may have on the results.

Section 5 – "The analysis did not reveal significant dependency of the start/end of the birch pollen season on the underlying pollen source map." – can you please explain why you would expect it to? I don't understand how this could be related to the source

map.

Technical corrections:

* Page 2 L10 – grammar – "skin prick test" -> "skin prick testing"

* Page 2 L18 - grammar - "The literature review presented in this section show" -> "The literature review presented in this section shows"

* Figure 2 – caption. Might be worth explaining what the units of % mean here (%) – I'm not sure all readers will understand this. I.e. 'percentage cover of birch in each 15km x 15km grid square'.

* Page 4 L16 – grammar – "The National Forest Inventory statistics was" -> "The National Forest Inventory statistics were"

* Page 5 L3 – typo - remove brackets from reference

* Page 5 L10 – typo – remove closing bracket.

* Figure 7 caption – typo – 'rght' to 'right'

* Page 9 L10 – grammar – "This suggest" -> "This suggests"

* Page 10 L18 – grammar – "advices" -> "advice"
* * *

---

## Referee Comment (RC3) · Anonymous Referee #3 · 21 Aug 2019

Review of "Incorporation of pollen data in source maps is vital for pollen dispersion models"

General comments:

This manuscript compares several different methods for forecasting atmospheric pollen concentrations, specifically for the case of birch pollen emissions and transport in Europe. The authors have performed pollen hindcast simulations in the Enviro-HIRLAM regional model, using three different source maps. The simulations were performed both with and without calibration using observed pollen data. Forecast skill is compared using objective metrics, including both traditional (continuous) metrics, and threshold-based skill metrics calculated from the hit rates for forecasts of daily mean pollen concentrations falling into four classes.

[Figure]

The study shows that calibration of pollen source maps using pollen observations significantly improved the model's performance on these standard metrics. Since the data used for the calibration of the model are the same data used to evaluate the model, this is unsurprising. The authors acknowledge that this is the main methodological limitation of the current study.

My main comment addresses this limitation. Specifically, the authors should consider whether a portion of the available data could be withheld from the calibration dataset, and used for evaluation. If this is not possible, they should more clearly explain why it cannot be accomplished. Also, the comment that "calibration uses annual or multi-annual average whereas the evaluation primarily concerned correlation and RMSE" should be clarified (e.g. by adding more details about the calculation of the metrics and calibration procedure).

In addition to this main comment, the authors should consider addressing the specific comments below. Additionally, the manuscript should be edited by a native speaker for English grammar and usage.

Specific comments:

1. p.5, l.19: Atmospheric models often assume aerosol shape factors and densities of 1 for simplicity; pollen grains can diverge significantly from this. Please comment on whether/how this is considered in the simulations described here. 2. p.5, l.26-27: please provide more detail about the correction for 2-m air temperature. 3. Description of SPIn method: The evaluation of maps calibrated with SPIn method plays a major role in this study. The method used for this map calibration is described in a cited paper (Prank et al., 2013). However, since evaluation of simulation results that use maps calibrated using this method is a central part of this manuscript, I think the calibration procedure should be described in a bit more detail. The method is mention on page 4 (lines 9-13), and there is a brief discussion of the method on page 5 (lines 30-34). A bit more detail should be added here to explain, briefly, how the ratio of modelled and

observed concentrations at 15 stations has been interpolated to create a ratio that is applied to emissions across the domain. 4. Presentation of metrics: the comparison of the different approaches using statistical metrics is an important aspect of this paper. Currently, the definitions of the acronyms are buried in the text, and the threshold-based metrics, which are likely less familiar to readers, are only available in referenced literature. I recommend briefly restating the definitions of the threshold-based metrics in a table or appendix. I also recommend summarizing the threshold value Cth and the ranges of concentrations for the low, medium, and high classes in a small table, so that readers can more easily reference this information. 5. Choice of metrics: The authors point out that the RMSE is highly sensitive to outliers (large discrepancies), which may limit its usefulness as a metric for this type of forecasting. Please consider whether metrics that have more recently come into use, such as the fractional absolute error (Yu et al., 2006), might be appropriate to use in addition to, or instead of, the mean bias and RMSE. 6. p. 8, l. 30: Clarify whether the bias is related to the general behaviors of the atmospheric model (e.g. the meteorology and simulated transport), or is a feature specifically of the parameterization of pollen flowering. 7. p. 9, l. 6-12 and l. 17-19, p. 10, l. 6-9: The authors attribute the remaining errors in pollen forecasts, after calibration of the pollen maps, to a need for additional improvement in the input datasets, and in the level of detail or calibration of the maps. I do not think that the results presented here are sufficient to support that conclusion. Other possible sources of error also need to be considered, and should be mentioned here (e.g., timing of pollen release, simulation of transport and removal processes). 8. Table 1 caption: Caption should be revised to include more information about how the metrics were calculated, in order to assist readers in interpreting the results. (e.g., metrics were calculated from daily mean modelled and observed pollen counts, using all available station data for both simulated domains, over X time period). Also, please clarify what the p-value refers to here. 9. Figures 2, 3, and 5: The red/green color bar used in Figure 2, and the color choices for the line plots in Figures 4 and 5, are probably not very colorblind-friendly. The authors may wish to consider choosing different color

schemes to make these figures accessible to more readers.

Technical corrections:

References: Yu, S. , Eder, B. , Dennis, R. , Chu, S. and Schwartz, S. E. (2006), New unbiased symmetric metrics for evaluation of air quality models. Atmosph. Sci. Lett., 7: 26-34. doi:10.1002/asl.125

---

## Author Comment (AC1) · 6 Sep 2019

**Author response to the referee comments to the paper by Kurganskiy et al.: Incorporation of pollen data in source maps is vital for pollen dispersion models**

We would like to thank all the reviewers for their comments and suggestions led to improving the paper. They are addressed below with our responses in blue font.

**Referee 1, Anonymous.**

It is my impression that the manuscript both falls within the scope, and is scientifically sound. The topic of the development of source maps, and the impact of these on the final model results, is very relevant for further development. However, this paper requires an intimate knowledge of pollen modelling and detailed reading of a number of background papers, to completely comprehend the analysis. The details of pollen dispersion modelling is not within my main area of expertise, and a thorough assessment of the method, is therefore beyond my current knowledge.

Response:
Thank you for the positive review. Specific questions/comments are addressed below. It is our impression that the combined requests from reviewer 1, 2 and 3 improve the manuscript and also address the above issues concerning prior knowledge on pollen dispersion modelling.

Questions / Comments

Do the authors have any comments on why the correlations are better for the stations in the P15 domain calculations?

Response:
  This is most likely related to the quality of the source maps (Fig 2) and which regions are affected more remote and large emission areas. Both Moscow and Finnish stations are regularly affected by long-range transport (LRT) from Russia both early and late in the season (e.g. Sofiev et al, 2006). This is also the case in our simulations where the model simulates large amounts of birch arriving from Northern sites late in the season (originally Fig. 4f and Fig. 4i, now Fig. 4g and Fig. 4j), which are not found in the observations. Contrary, Denmark is regularly affected by local sources and transport from Poland and Germany early in the season (Skjoth et al, 2007). In our simulations model domain T15 contains the large emission source in parts of Russia in all three maps (Fig. 2), which unfortunately do not have a nearby calibration stations. This emission source is not present in the model domain P15. This suggests, that the model overestimates the LRT component originating from this area, where a likely reason for this is related to the strength of the pollen source. In order to clarify this we have modified this sentence to the manuscript page 8, line 32-33 from:
".. runs, which complements a multi-model ensemble, that demonstrated generally good results for Europe but also highlighted .."
Into
".. runs. One of the possible reasons for this is that the large pollen emission area found in Russia, North of Moscow and Smolensk is overestimated for this particular year. The source is found in all three source maps and the overestimation is further supported by the fact that the model simulates large amounts of birch pollen concentration very late in the season at Moscow and Smolensk (Fig 4g and Fig 4j), most likely originating from more Northerly regions. This area will occasionally also affect the Finnish stations as it is well known that this region receives LRT from Russia (e.g. Sofiev et al, 2006). The study therefore complements a multi-model ensemble, that demonstrated generally good results for Europe which also highlighted."

It is interesting that the "moderate" concentrations is generally underestimated and the "high" is generally overestimated by all combinations of model/maps. Could the authors elaborate on possible explanations for this?

Response:
A large number of the simulated high concentrations are found in Finland and Russia and could be caused by the aforementioned overestimation of the large birch pollen source found in Russia. An underestimation of moderate concentrations can be caused by the fact that all three maps will not be able to identify sources with a small geographical extent as they cannot be reproduced by the land cover data set. This will potentially increase pollen emissions with a small amount in many regions, which typically will only have a local influence, contrasted the dense emission areas like parts of Russia, that will also have a significant effect on long distance transport episodes. In order to elaborate more on this this we have added this sentence to the manuscript page 9, line 10 after Map 2.:
"The small-scale pattern in birch habitation will be difficult to capture by the land cover data sets, which will have the effect that a small but diffuse emission source found in many locations is not present, causing the model to underestimate medium concentration levels. Contrary the large emission source found in parts of Russia will cause the model to overestimate periods, when long distance transport is present. This effect of over and underestimation will vary from year to year, depending on pollen productivity which is known to vary from year to year and in between regions (Ranta et al, 2005)."

What do the authors think is the explanation for the variation in the scaling factor, and the high values near Copenhagen (Discussion, page 8, line 26)?
Is this map relevant for other applications, e.g. for selecting areas where further studies are needed to establish why modelled and observed values do not agree?

Response:
The variation in the scaling factor shows variation of pollen production, which is known to vary substantially from region to region. Copenhagen receives a substantial fraction of pollen from sources found within the city itself and only occasional LRT (Skjoth et a, 2007, 2008b, Mahura et al, 2007). It has previously been shown that urban areas enhance pollen production such as ragweed (e.g. Ziska et al, 2003). This might also be the case for birch, which will have the consequence that the scaling factor will be large in Copenhagen compared to other less urbanised regions or regions where a large fraction of the pollen may be due to LRT such as Moscow. In order to clarify this we have added this sentence to the manuscript on page 8, line 29, and the reference by (Ziska et al, 2003)
"Add to this, that urban areas have previously been identified as areas that may enhance pollen production of flowering plants (Ziska et al., 2003). This may be particular relevant for Copenhagen as a substantial fraction of the birch pollen observed is expected to originate from a source found in the city itself (Skjoth et al, 2008b)."

Page 7, line 14-15 / 20-21 – The Danish sites are mentioned as having the highest R2. Could you comment on why the SCF run appears to result in lower R2 than the COR at these sites, opposite of most other sites. Perhaps in the discussion section.

Response:
The reviewer has identified a section that needs slightly more explanation. We therefore suggest to add following sentences to page 7, line 15.
"It should here be noted that the $R^2$ values decrease slightly at the Danish sites according to the SCF runs. The reason for this is that the simulated concentrations in the SFC runs are increased on almost every single day, which has the effect of reducing the underestimation found in the

central season (e.g. Fig. 5a) and increasing LRT, most likely originating from sources found in Germany and Poland in the beginning of the season and from Scandinavia in the end of the season."

Minor comments and/or typographical corrections

Page 5, line 7 - "∶ ∶ ∶ f or use to simulate of the birch flowering season". I suggest to delete "of".

Response:
Thank you. It's corrected.

Page 5, line 10 – Reference for details on the choice of the seasonal pollen productivity?

Response:
The value has been taken from the SILAM model code using the source term described in Sofiev et al, 2013, which has been added to the line. Note that other values were used in the multi-model ensemble by Sofiev et al (2015) and in the EMPOL model used in COSMO-Art (Zink et al, 2013), suggesting that such values may be updated from time to time based on the latest scientific developments in atmospheric modelling.

Page 5, line 28 – Could you add further description of the pollen emissions scaling factor? It is not clear whether it is this factor that is further explained in line 30 and onward? If so, please apply the acronym here (line 30). Also, in line 30-31, it is unclear whether the scaling factor is based on the COR-run (simulations of type 2), and then following applied for all maps?

Response:
The pollen emissions scaling factor is the same as explained in line 30 and onward.
The following set of changes has been added to the manuscript for clarification:

1. Line 30: "The scaling factor…" has been changed to "The scaling factor for pollen emissions"
2. Line 31: The acronym has been added: "…from the simulations of type 2 (COR)"
3. Line 33: "….thus providing 6 grid based scaling factors (Fig. 3)." has been changed to ""….thus providing 6 grid based scaling factors (Fig. 3) applied for the SCF simulations for all maps."

Page 5, line 34 - "∶ ∶ ∶ the COR ones∶ ∶ ∶" I suggest to change to "COR-runs".

Response:
Thank you. It's been changed.

Page 5, line 34 - Two closing-parenthesis, delete one.

Response:
Thank you. It's corrected.

Page 6, line 1 – Again a reference to "the grid based scaling factor". I suggest to consistently use the abbreviation, you introduced on page 5, if it is the same "grid based scaling factor".

Response:
Thank you. It is the same grid based scaling factor. The following correction has been added:

Page 6, L1: "the grid based scaling factor (shown in Fig. 3)" has been changed to "the grid based scaling factor (obtained from the COR runs and shown in Fig. 3)".

Page 6, line 14 – Define LRT-abbreviation as Long Range Transport.

Response:
Thank you. The abbreviation is defined.

Page 7, line 14 – I suggest to add a reference to appendix A for the referred MB results, already in this line. I am aware that the reference is listed in the final line of this section, but I would prefer it also listed when the results are first mentioned.

Response:
Thank you. The reference is added.

Page 8, line 9 – HR has already been defined as Hit rate (page 6, line 28), so there is no need to write it out here.

Response:
Thank you. It's corrected.

Page 8, line 30 – "(too late flowering end")". I suggest to rephrase, e.g. "delay of end of flowering".

Response:
Thank you. It's rephrased as suggested.

Page 9, line 22 - ":::the need local calibrations:::.". Please add "for": ":::the need for local calibrations."

Response:
Thank you. It's added.

Page 9, line 28-29 – This information borders on "method", could the latter part be moved to methods?

The latter part (i.e., "...the evaluation primarily concerned correlation and RMSE.") has been added to the methods section (Page 6., L17):
"...error (RMSE)" -> "..error (RMSE) and the evaluation primarily concerns correlation and RMSE."

Figures

Fig 1 – I suggest to add a scale bar.

Response:
A scale bare has been added as suggested

Fig 4-5. What is the order of the charts based on? The first and last appears to be the P15 domain, and the rest, the T15 domain. Perhaps move (a) and (l) next to each other, if there are no other reason behind the order.

Response:

There is no any specific reason for the ordering. As suggested, Figures (a) and (I) have been moved next to each other and the updated Figures 4-5 have been provided.

**Referee 2, Rachel McInnes**

General comments: This is a really nice piece of work which adds to a very topical area of study. It is novel and extends scientific progress in this area. I believe it is relevant to the scope of the Atmospheric Chemistry and Physics journal.
The authors demonstrate the importance of including observational pollen data in producing pollen source maps for pollen dispersion modelling, and they outline the health impacts for the population which is why this modelling is so important.

Response:
Thank you for the positive review and specific questions/comments which are addressed below.

Specific comments:

Section 2.1 – while you have provided information on the datasets used and references to explain the methodological differences between map 1, map 2 and map 3 – it is quite difficult to overview the 3 methods at a glance, in order to compare and contrast. I suggest that a graphical diagram outlining the method / flowchart for each different map might help the reader understand the differences between the three maps being compared.

Response:
Thank you, this is a very good suggestion! The suggested flowcharts are produced and they have been added to the paper (see the appendix section) with corresponding reference (P5, L27):
"…and forest plot data." -> "…and forest plot data (see Fig. A1 in the appendix section)."

[Figure]

Figure A1. Flowcharts outlining the data and methods used to obtain Map 1 (a), Map 2 (b) and Map (3).

Section 2.2& 2.3 – what dates were the models run over? This needs clarified please.
E.g. in Section 2.2 when discussion the GDD data - can you clarify what dates were used for the input data? In 2.3 you state: "Input meteorological initial and boundary conditions were taken from the ECMWF-IFS model (Persson, 2011) with 15 km resolution and 6-hour interval" – this would be a good point to say what years the models were run over. When you mention the Birch 2006 season – is that this work, or are you referring to previous work?

Response:
The Enviro-HIRLAM model has been run for the period starting from 1st March until 15 of June, 2006. GDD is an integral part of Enviro-HIRLAM and GDD calculations are performed for the same dates as the model is run over. The ECMWF-IFS data have been extracted from the ECMWF archive for the same dates (1st March – 15 June). The starting date (1st March) has been selected as the reference date used to obtain the temperature sum threshold maps utilized as input data in the study. The Birch 2006 season refers to this work. The following changes are added:

1. Page 5 L8. Clarification sentence. "GDD is an integral part of Enviro-HIRLAM and GDD calculations are performed for the same dates as the model is run over."
2. Page 5 L25. "…has been run for the birch pollen season 2006." has been changed to "…has been run from 1st March until 15 June, 2006 with the ECMWF input data covering the same period."

Section 2.3 – what was the reason for excluding the method 1 results from the results section?

Response:
The reason was simply to avoid overloading of the reader. Temperature bias correction (method 2) is a standard approach in the pollen dispersion model application and in this particular study it mainly affects the start of the season (when comparing the results of method 1 and 2). Therefore, method 2 was chosen as a reference point in the paper.

Section 3 – "The largest differences between the modelled and observed number of cases are found for the moderate and high pollen concentrations." – but are these all within the error bars, see Fig 7?

Response:
The reviewer has a good point as this needs clarification. We have therefore implemented this: Page 8, L5. The sentence "The largest differences between the modelled and observed number of cases are found for the moderate and high pollen concentrations." has been changed to "The largest differences between the modelled and observed number of cases are found for the moderate and high pollen concentrations, but the differences are within the error bars (see Fig.7)".

Section 4 – can you comment in the manuscript on what work you believe would need to be done to try to improve the modelling of the last flowering day?

The following sentence has been added to section 4 (Page 8, L32): "Introducing more pollen observation stations and performing more iterations for rescaling of pollen emissions (i.e. repeating SCF run several times) would strengthen the current approach and it could potentially improve the modelling of the last flowering day." Add to this that certain areas with numerical but geographical small source areas need be identified as they cannot be resolved by current land cover data sets. This aspect is now covered in more detail in the expanded discussion – see the response to reviewer 1.

Section 4 – "that also demonstrated the need for recalibration of the source term. However, it has since been shown by Zink et al. (2017) that source terms combining pollen data from several years with detailed land cover data can outperform other approaches minimizing the need local calibrations. " - Linked to my earlier question - has this study only been carried out for the one year? Please clarify and comment on the impact of how long a period you are doing this analysis over may have on the results.

Response
The study by Zink et al (2017) was only covering one season. The study by Sofiev 2017 that provided data for map 2 covered many years. Nevertheless, the annual variations are large and substantial improvements can be obtained by calibrating long-term pollen based maps with the data from the year that is being studies using data fusion. This aspect is now more clear in the expanded discussion – see the response to reviewer 1, in particular the extension on page 9, line 10 after Map 2 as the new paragraph as stronger foundation behind the recommendation to use data fusion.

Section 5 – "The analysis did not reveal significant dependency of the start/end of the birch pollen season on the underlying pollen source map." – can you please explain why you would expect it to? I don't understand how this could be related to the source map.

Response:
We did not expect to see a big difference between the start/end dates (on average over the stations) using different pollen source maps. However, the birch source maps are involved in calculations of the pollen emissions and, consequently, concentrations. The start and end of the season were calculated as the dates when the accumulated pollen concentrations reach 5% and 95% of the annual pollen sums (Annual Pollen Integral). Therefore, the starting/ending dates simulated with the maps were inter-compared, but not included in the analysis in details. Add to this that we have explained the effect of the large Russian source in more detail within the manuscript and its effect on LRT in Moscow and Smolensk. Unfortunately, this large region has no pollen observing sites centrally in the emission area so it will be very difficult to rescale the emission in that area without using more advanced approaches that takes into account the foot print area of the observations or ideally adding more calibration sites as stated both in the discussion and the conclusion. In order to clarify this we have modified a sentence on page 9 line 18 from:
" .. introducing more observational points and/or"
Into
".. introducing more observational points, in particular in areas sparse with data or regions with high emissions such as parts of Russia and/or .."

Technical corrections:

* Page 2 L10 – grammar – "skin prick test" -> "skin prick testing"

Response:
Thank you. It's corrected.

* Page 2 L18 - grammar - "The literature review presented in this section show" -> "The literature review presented in this section shows"

Response:
Thank you. It's corrected.

* Figure 2 – caption. Might be worth explaining what the units of % mean here (%)
– I'm not sure all readers will understand this. I.e. 'percentage cover of birch in each

15km x 15km grid square'.

Response:
Thank you. The clarification has been added.

* Page 4 L16 – grammar – "The National Forest Inventory statistics was" -> "The National Forest Inventory statistics were"

Response:
Thank you. It's corrected.

* Page 5 L3 – typo - remove brackets from reference

Response:
Thank you. It's corrected.

* Page 5 L10 – typo – remove closing bracket.

Response:
Thank you. It's corrected.

* Figure 7 caption – typo – 'rght' to 'right'

Response:
Thank you. It's corrected.

* Page 9 L10 – grammar – "This suggest" -> "This suggests"

Response:
Thank you. It's corrected.

* Page 10 L18 – grammar – "advices" -> "advice"

Response:
Thank you. It's corrected.

**Referee 3, Anonymous**

General comments:

This manuscript compares several different methods for forecasting atmospheric pollen concentrations, specifically for the case of birch pollen emissions and transport in Europe. The authors have performed pollen hindcast simulations in the Enviro-HIRLAM regional model, using three different source maps. The simulations were performed both with and without calibration using observed pollen data. Forecast skill is compared using objective metrics, including both traditional (continuous) metrics, and threshold-based skill metrics calculated from the hit rates for forecasts of daily mean pollen concentrations falling into four classes.

The study shows that calibration of pollen source maps using pollen observations significantly improved the model's performance on these standard metrics. Since the data used for the calibration of the model are the same data used to evaluate the model, this is unsurprising. The authors acknowledge that this is the main methodological limitation of the current study.

My main comment addresses this limitation. Specifically, the authors should consider whether a portion of the available data could be withheld from the calibration dataset, and used for evaluation. If this is not possible, they should more clearly explain why it cannot be accomplished.

Response:
The reviewer is right on this aspect and clarification is needed. It should here be noted, that a central conclusion from this study is as following:
"This suggest, that accurate exposure calculations that use dispersion models preferably should use data fusion that combine a detailed inventory based source term", which is both directly related to the title of the study and the main purpose of the study. Removing a fraction of the data will remove the foundation for this conclusion. It is our impression that the reviewer expected this as he writes that it might not be possible to remove a fraction of the observations to act as an independent data set. Having said this, then the exercise without using any of the calibration data has already been carried out through the scenarios called COR. As such a full independent evaluation is available. The fact that the model are heavily dependent on data in order to produce good results, where the same data is also needed for validation is a classical dilemma in science. Traditionally this dilemma is solved by a procedure called cross validation, e.g. where one data point is removed in the calibration and then the model is rerun without that calibration point in order to simulate the value in the calibration point. The procedure is then repeated until all calibration points have been simulated. In our case this would mean 15 additional simulations for each map and modelling domain and the computation costs for this makes this unfeasible. It should here be noted that this cross validation is in practice just a sensitivity study and that it will not affect the conclusion. So, omitting this part will not affect the quality of the study.

In order to clarify that a full independent evaluation is present in this study and that a further investigation of the sensitivity of the calibration points on the model results (e.g. using cross validation) would require an unfeasible amount of extra calculations, we therefore suggest the following changes to the manuscript:

Page 9 L29: "It should be noted, that an independent evaluation that do not use pollen data is shown by analysis of the COR runs in the study. Another approach could be to carry out a cross-validation procedure using the so-called leave one out procedure as this tests the sensitivity of individual data points. However, such sensitivity study requires 15 additional model simulations for each map and each modelling domain, hence 90 simulations. The increased computational cost makes this exercise unfeasible suggesting that other approaches needs to be developed. "

Also, the comment that "calibration uses annual or multiannual average whereas the evaluation primarily concerned correlation and RMSE" should be clarified (e.g. by adding more details about the calculation of the metrics and calibration procedure).

Response:
Thank you. Clarification of the metrics calculation and calibration procedure is added. Details and specific changes are found in the response to comments 3, 4 and 8, which are addressed below.

In addition to this main comment, the authors should consider addressing the specific comments below. Additionally, the manuscript should be edited by a native speaker for English grammar and usage.

Thank you. The specific comments are addressed below. The updated version of the manuscript contains the grammar corrections provided by referee 1 and referee 2, which includes a native

English speaker. Add to this that the manuscript has been thoroughly checked both manually and using dedicated software for adjusting the grammar. Minor additional changes to the manuscript are found in a specific list below termed additional improvements to the grammar.

Specific comments:

1. p.5, l.19: Atmospheric models often assume aerosol shape factors and densities of 1 for simplicity; pollen grains can diverge significantly from this. Please comment on whether/how this is considered in the simulations described here.

Response:
Pollen grains are considered as near spherical aerosols with an estimated particle size 22 um and density 800 kg/m$^3$ (Sofiev et al., 2006). The following change has been added to clarify this:

 Page 5, L17. "…and an estimated size of 22 um (e.g. Mäkelä, 1996)." has been changed to ""…and an estimated size of 22 um (e.g. Mäkelä, 1996) with near spherical shape (Sofiev et al., 2006)."

2. p.5, l.26-27: please provide more detail about the correction for 2-m air temperature.

Response:
The following details to the manuscript are added for clarification:
Page 5, L.30, after "...the results section.": "The correction for $T_{2m}$ bias is done in two steps. Firstly, the model output from the simulations of type 1 is used to calculate the biases (differences between simulated and assimilated $T_{2m}$) for each assimilation window (i.e. 6 hours). Secondly, the calculated biases are introduced in the model simulations of type 2 (COR) where the simulated $T_{2m}$ is corrected at each model time step using the average bias between 2 nearest/closest assimilation windows. Further details of this procedure can be found in Kurganskiy (2017)."

3. Description of SPIn method: The evaluation of maps calibrated with SPIn method plays a major role in this study. The method used for this map calibration is described in a cited paper (Prank et al., 2013). However, since evaluation of simulation results that use maps calibrated using this method is a central part of this manuscript, I think the calibration procedure should be described in a bit more detail. The method is mention on page 4 (lines 9-13), and there is a brief discussion of the method on page 5 (lines 30-34). A bit more detail should be added here to explain, briefly, how the ratio of modelled and observed concentrations at 15 stations has been interpolated to create a ratio that is applied to emissions across the domain.

Response:
We agree with the reviewer and the following clarification is added to the manuscript on page 5, L32:
"The interpolation procedure takes into account the weighted distance between each observation point and grid cell with a constant radius of influence (1 km in this study). The procedure also ensures the scaling factors are equal to around 1 in the areas located far away from the observation points. This is especially visible in P15 domain, e.g. in France (Fig. 3 a,c,e). Further details of the interpolation procedure can be found in Kurganskiy (2017)."

4. Presentation of metrics: the comparison of the different approaches using statistical metrics is an important aspect of this paper. Currently, the definitions of the acronyms are buried in the text, and the threshold-based metrics, which are likely less familiar to readers, are only available in referenced literature. I recommend briefly restating the definitions of the threshold-based metrics in a table or appendix. I also recommend summarizing the threshold value Cth

and the ranges of concentrations for the low, medium, and high classes in a small table, so that readers can more easily reference this information.

Response:
Thank you! We have implemented the suggestions proposed by the reviewer. The definitions of the threshold-based metrics are added to the appendix section (see Table A6). The threshold value Cth and the ranges of concentrations are summarized in a table and added to the appendix section as well (see Table A5). The following sentence is added in the text:
Page 6., L. 29 (end of the methods section): "For summary of the thresholds, classes and metrics used to calculate the threshold-based statistics the reader is referred to Tables A5 – A6 in the appendix section."

5. Choice of metrics: The authors point out that the RMSE is highly sensitive to outliers (large discrepancies), which may limit its usefulness as a metric for this type of forecasting. Please consider whether metrics that have more recently come into use, such as the fractional absolute error (Yu et al., 2006), might be appropriate to use in addition to, or instead of, the mean bias and RMSE.

Response:
Two additional metrics has been added to the analysis: Normalized Mean Bias Factor (NMBF) and Normalized Mean Absolute Error Factor (NMAEF) as the robust measures recommended by Yu et al., 2006. NMBF and NMAEF values can be found in Table 1 as well as Tables A1-A4 in the appendix section. Introducing the new metrics led to the following changes in the text:
Page 6, L.17. We added the following sentence: "Two additional metrics Normalized Mean Bias Factor (NMBF) and Normalized Mean Absolute Error Factor (NMAEF) are also calculated according to Yu et al., (2006}) since RMSE is highly sensitive to outliers.
Page 7, L.13-14: "...but it reduced the mean bias (MB)..." was changed to "...but it reduced the mean bias (MB, NMBF)..."
Page 7., L.18-19: "and a lower mean bias (MB = 69.08 pollen m$^{-3}$)..." was changed to and a lower mean bias (MB = 69.08 pollen m$^{-3}$, NMBF = 0.09 )...".
Page 7, L.22: "... and decreasing the MB values ..." was changed to "... and decreasing the MB, NMBF, NMAEF values ...".

6. p. 8, l. 30: Clarify whether the bias is related to the general behaviors of the atmospheric model (e.g. the meteorology and simulated transport), or is a feature specifically of the parameterization of pollen flowering.

Response:
The study suggests that the bias is related to the overestimated long-range transport caused by the areas in Russia where no calibration points are available. See our response to reviewer 1, 2 with corresponding changes added to Page 8, L32.

7. p. 9, l. 6- 12 and l. 17-19, p. 10, l. 6-9: The authors attribute the remaining errors in pollen forecasts, after calibration of the pollen maps, to a need for additional improvement in the input datasets, and in the level of detail or calibration of the maps. I do not think that the results presented here are sufficient to support that conclusion. Other possible sources of error also need to be considered, and should be mentioned here (e.g., timing of pollen release, simulation of transport and removal processes).

Response:
The reviewer has a good point and we have therefore made the following change to the manuscript on Page 9, L12.
"Other possible sources of error (e.g., timing of pollen release, quality of meteorological parameters, simulation of atmospheric transport and removal processes) could also be taken

into account. However, their effect is less important in comparison with the quality of emission (source) maps – the main uncertainty in air pollution and pollen dispersion modelling."

8. Table 1 caption: Caption should be revised to include more information about how the metrics were calculated, in order to assist readers in interpreting the results. (e.g., metrics were calculated from daily mean modelled and observed pollen counts, using all available station data for both simulated domains, over X time period). Also, please clarify what the p-value refers to here.

Response:
Thank you. The caption has been revised as suggested. The p-value < 0.01 for all maps. This is also specified in the caption.

9. Figures 2, 3, and 5: The red/green color bar used in Figure 2, and the color choices for the line plots in Figures 4 and 5, are probably not very colorblind-friendly. The authors may wish to consider choosing different color schemes to make these figures accessible to more readers.

Response:
We agree that the colours are not very colorblind-friendly. It is especially relevant for Figures 4-5. Therefore, the observed time series (Figures 4-5) are now depicted in "black" instead of "red" in order to make these figures accessible to more readers. The line types were changed to lines with points.

Technical corrections:
References: Yu, S. , Eder, B. , Dennis, R. , Chu, S. and Schwartz, S. E. (2006), New unbiased symmetric metrics for evaluation of air quality models. Atmosph. Sci. Lett., 7: 26-34. doi:10.1002/asl.125

Thank you for providing the references. It has been added to the paper together with the requested statistical metrics.

Additional improvements to the grammar

Page 6, L.9: "...each stations..." -> "...each station..."
Page 9, L.12: "...pollen concentration..." -> "...pollen concentrations..."
Page 9, L.13: "..relatively to.." -> "...relative to..."

References

Mahura, A. G., Korsholm, U. S., Baklanov, A. A., and Rasmussen, A.: Elevated birch pollen episodes in Denmark: contributions from remote sources, Aerobiologia, 23, 171–179, https://doi.org/10.1007/s10453-007-9061-3, 2007.

Ranta, H., Oksanen, A., Hokkanen, T. et al. Int J Biometeorol (2005) 49: 146. https://doi.org/10.1007/s00484-004-0228-0

Skjøth, C. A., Sommer, J., Stach, A., Smith, M., and Brandt, J.: The long-range transport of birch (Betula) pollen from Poland and Germany causes significant pre-season concentrations in Denmark, 37, 1204–1212, https://doi.org/10.1111/j.1365-2222.2007.02771.x, 2007.

Skjøth, C. A., Sommer, J., Brandt, J., Hvidberg, M., Geels, C., Hansen, K., Hertel, O., Frohn, L., and Christensen, J.: Copenhagen – a significant source of birch (Betula ) pollen?, Int. J. Biometeorol., 52, 453–462, 2008b.

Sofiev, M., Siljamo, P., Ranta, H., and Rantio-Lehtimäki, A.: Towards numerical forecasting of long-range air transport of birch pollen: theoretical considerations and a feasibility study, International Journal of Biometeorology, 50, 392–402, https://doi.org/10.1007/s00484-006-0027-x, 2006.

Ziska LH, Gebhard DE, Frenz DA, Faulkner S, Singer BD, Straka JG (2003) Cities as harbingers of climate change: common ragweed, urbanization, and public health. J Allergy Clin Immunol 111:290–295.

---

## Author Response (AR2)

**Author response to the referee comments to the paper by Kurganskiy et al.: Incorporation of pollen data in source maps is vital for pollen dispersion models**

We would like to thank anonymous referee 3 for comments and suggestions led to improving the paper. They are addressed below with our responses in blue font.

**Referee 3, Anonymous.**

Dear authors,

Before the paper can be finally published, please carefully revise the manuscript according the referee's comments as follows:

1) in the Table 1 caption, it has not yet been clarified what the p-value means here. I think it's important to clarify which test was performed and which things were being compared, for example with a sentence like "The differences with the observations were statistically significant for all maps (p < 0.01 by a Student's t-test)"... or something similar.

Thank you! We have added the following sentence in the Table 1 caption:
"The results were statistically significant for all maps (p-value < 0.01 according to a Student's t-test)."

2) In Table A6, I found the definition of the Odds Ratio (OR) difficult to understand. I recommend it be rephrased for clarity.

Thank you! We have changed the OR definition from
"OR indicates how much higher are the chances to get pollen concentration > Cth than < Cth if the model prediction is > Cth."
into
"OR shows the likelihood to get concentrations > Cth compared to concentrations < Cth when the model prediction is > Cth, hence POD/POFD"